# Air-Decoding: Attribute Distribution Reconstruction for Decoding-Time Controllable Text Generation

**Tianqi Zhong[1], Quan Wang[2], Jingxuan Han[1], Yongdong Zhang[1], Zhendong Mao[1]***

[1]University of Science and Technology of China
[2]MOE Key Laboratory of Trustworthy Distributed Computing and Service,
Beijing University of Posts and Telecommunications
{ztq602656097, hjx999222}@mail.ustc.edu.cn
wangquan@bupt.edu.cn, {zhyd73, zdmao}@ustc.edu.cn

## Abstract

Controllable text generation (CTG) aims to generate text with desired attributes, and decoding-time-based methods have shown promising performance on this task. However, in this paper, we identify the phenomenon of Attribute Collapse for the first time. It causes the fluency of generated text to rapidly decrease when the control strength exceeds a critical value, rendering the text completely unusable. This limitation hinders the effectiveness of decoding methods in achieving high levels of controllability. To address this problem, we propose a novel lightweight decoding framework named Air-Decoding. Its main idea is reconstructing the attribute distributions to balance the weights between attribute words and non-attribute words to generate more fluent text. Specifically, we train prefixes by prefix-tuning to obtain attribute distributions. Then we design a novel attribute distribution reconstruction method to balance the obtained distributions and use the reconstructed distributions to guide language models for generation, effectively avoiding the issue of Attribute Collapse. Experiments on multiple CTG tasks prove that our method achieves a new state-of-the-art control performance[1].

## 1 Introduction

Controllable text generation (CTG) aims to produce texts with specific attributes(e.g. sentiment, topic, non-toxicity). In this field, mainstream methods using Transformer-based Casual Language Models (CLMs) like GPT-2 (Radford et al., 2019) and GPT-3 (Brown et al., 2020) have achieved fairly good attribute control. Even so, it remains challenging to control the generated text to simultaneously satisfy certain attributes and maintain a reasonable coherence (Carlsson et al., 2022).

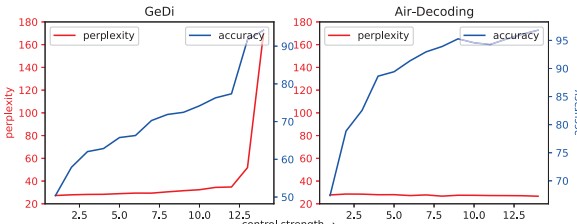

Figure 1: The phenomenon of Attribute Collapse (left) and the elimination of Attribute Collapse (right), which are both based on the sentiment control task. The perplexity is an inversely proportional metric to fluency and the accuracy is a directly proportional metric to attribute relevance.

Current CTG methods can be divided into three categories (Zhang et al., 2022). The first category achieves attribute controllability by retraining the whole parameters of CLMs (Keskar et al., 2019; Wang et al., 2021). These methods obtain impressive control effects, but as the scale of CLMs increases, the computational cost becomes excessively large. The second category fine-tunes prefixes or prompts to control CLMs in generating texts with specified attributes (Yu et al., 2021; Zhang and Song, 2022; Qian et al., 2022). These methods have low computational cost and fast generation speed, but the control prefixes or control prompts absorb the features of the training corpus and are prone to overfitting, resulting in poor generality of the generated text. The third category is the decoding-time approach, which guides the model to generate the desired attribute text by adjusting the output probability distribution of the model during the decoding stage. For example, most methods use an attribute distribution obtained by a well-trained classifier to control the CLMs' output distribution and add an exponential term to the attribute distribution as a control strength (Liu et al., 2021; Yang and Klein, 2021; Krause et al., 2021). Due to no direct fine-tuning of the models used for the generation, these methods exhibit re-

---

*Corresponding author: Zhendong Mao.

[1]The code implementation is available at https://github.com/R1047/Air-Decoding.

markable generalization performance and are easy to achieve high control effectiveness when increasing control strength, which is hardly achievable by the first two methods.

However, decoding-time approaches suffer from **Attribute Collapse**, which refers to the phenomenon that when the control strength increases to a certain critical value, the fluency of the generated text will rapidly decrease. As illustrated in Figure 1(left), where the fluency of generated texts would severely deteriorate as the control strength increases. We found the reason lies in the imbalanced attribute distribution, where the weights of attribute words are significantly higher than those of non-attribute words. As a result, when given a high control strength, the final output distribution will deviate from the original language model's output distribution and excessively favor the attribute distribution. This leads to the generated tokens favoring specified attributes but ignoring basic semantics and syntax, resulting in generated text with high attribute accuracy but compromised fluency.

We propose **Air-Decoding**, a novel lightweight decoding framework to address the issue of Attribute Collapse in traditional decoding-time approaches. Our method could obtain a more balanced attribute distribution in a low-resource manner, which helps us achieve better generation results while reducing resource consumption. Concretely, We first use a lightweight approach to obtain corresponding conditional language models for each attribute, which serves as our attribute classifier to generate the original word-level attribute distribution. Then, we propose an **Attribute Distribution Reconstruction** method to reconstruct the original attribute distribution, which avoids exaggerating the weights of attribute words and excessively diminishing the weight of non-attribute words, resulting in a more balanced attribute distribution. As shown in Figure 1(right), after applying our Air-Decoding framework, the perplexity could remain within a stable range as the control strength increases, proving that generated texts based on the reconstructed attribute distribution ensure high attribute accuracy and good semantic and syntactic structures.

Our main contributions are as follows:

- We first identify the phenomenon of Attribute Collapse in the CTG tasks and develop a novel lightweight decoding framework named Air-Decoding specifically to address this issue. In the framework, we designed an attribute distribution reconstruction method to obtain a more balanced attribute distribution, which results in better CTG performance.

- We introduce prefix-tuning to obtain attribute classifiers, which enables us to achieve the attribute distribution with a low-resource approach.

- We conduct experiments on three CTG tasks: sentiment control, topic control, and text detoxification. The results prove that our Air-Decoding method achieves a new **SOTA** control performance. In particular, while achieving an improvement in attribute accuracy, we also attain a substantial enhancement in fluency.

## 2 Related Work

In recent years, numerous controllable text generation methods based on pre-trained language models have emerged (Yang et al., 2022; Madotto et al., 2020; Ziegler et al., 2019). These methods can be roughly categorized into three types: Retrain/Refact, Prefix/Prompt-tuning, and Decoding-time approach (Zhang et al., 2022).

**Retrain, Refactor**. Retraining and refactoring aim to retrain a conditional language model from scratch or change its architecture. Keskar et al. (2019) use 55 attribute control codes to finetune a 1.63 billion-parameter transformer to control generation. Chan et al. (2020) added a control block to GPT-2's architecture and retrained the model using self-supervised learning methods with special control codes. Zhang et al. (2020) introduce POINTER, a modified Transformer model for lexically constrained text generation.

**Prefix/Prompt-tuning**. With the expansion of pre-trained language models, there is increasing interest in lightweight fine-tuning methods such as prefix-tuning (Li and Liang, 2021) and prompt-tuning (Lester et al., 2021). Qian et al. (2022) introduce contrastive prefixes that consider inter-prefix relationships during multiple prefix training. Zhang and Song (2022) add an attribute discriminator to control-prompt training using the unlikelihood method, but this increases training time significantly. Gu et al. (2022b) train an auto-encoder to encode training set samples into prefixes to guide GPT-2, achieving impressive controllability but suffering from poor diversity and generality.

**Decoding-time**. The decoding-time method is another approach to achieve CTG and its main idea

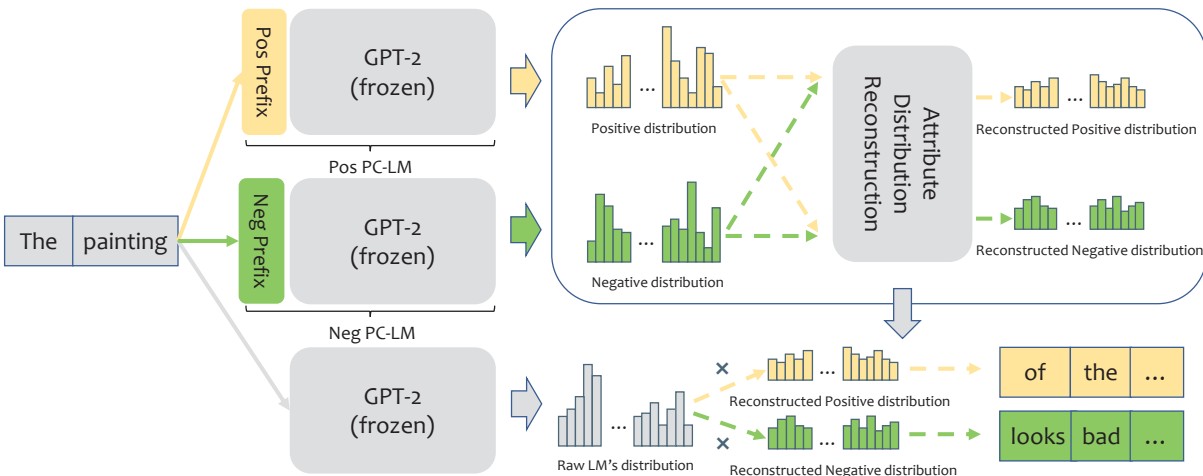

Figure 2: An illustration of Air-Decoding. Given the prompt "The painting", the pos PC-LM, neg PC-LM, and raw GPT-2 generate positive, negative, and raw distribution. Then, we use a reconstruction method to make these attribute distributions more balanced. Finally, we use the reconstructed attribute distributions as weights for the raw distribution to generate the next token.

is to adjust the probability distribution of the language models' output during the decoding stage. PPLM (Dathathri et al., 2019) utilizes the results of a pre-trained classifier through backpropagation to update the model's hidden states, thereby steering the hidden states toward generating text with the specified attributes. Yang and Klein (2021) directly use a classifier to calculate the probability of generating the next token in the sequence. Liu et al. (2021) uses an expert and an anti-expert to guide GPT-2 generating but this method may not be well-suited for multi-category attribute-controlled generation. GeDi (Krause et al., 2021) employs a class-conditional language model(CC-LM) as generative discriminators to direct the text generation using a base GPT-2 model. Lin and Riedl (2021) enhance GeDi for controllable story generation by incorporating a planning module in their Plug-and-Blend approach. Gu et al. (2022a) propose a lightweight regulator to adjust control strength at different decoding positions but it works not well under high-intensity control conditions.

## 3 Methodology

We propose Air-Decoding, a decoding-time-based CTG framework. Its main idea is reconstructing the attribute distributions to make the weights of attribute words and non-attribute words more balanced to generate more fluent text. Specifically, we propose Prefix-Conditional LM (PC-LM) which employs prefix-tuning to enable GPT-2 to acquire attribute distributions. Then, we design an attribute

distribution reconstruction method to balance the obtained distributions, which will be used to guide a frozen GPT-2. The overall framework is illustrated in Figure 2.

### 3.1 Preliminary

Controllable text generation aims to guide an autoregressive model $\mathcal{G}$ (i.e., GPT2) to generate texts with desired attribute. In this paper, we mainly introduce our method through the sentiment control task that includes positive and negative attributes for ease of understanding. However, our method exhibits generalizability and can be applied to other CTG tasks. Concretely, we usually give a prompt $x_{1:T-1} = \{x_1, x_2, \cdots, x_{T-1}\}$ (such as "The painting" in Figure 2) and a desired attribute $a$ (e.g. positive), and ask the model to generate the continuations $x_{T:N} = \{x_T, x_{T+1}, \cdots, x_N\}$, ensuring the whole text $x_{1:N}$ satisfies the given attribute. It can be formulated as:

$$P(x_{T:N}|x_{1:T-1}, a) = \Pi_{t=T}^{N} P(x_t|x_{<t}, a) \quad (1)$$

Thus, we need to model $P(x_t|x_{<t}, a)$ to achieve $P(x_{T:N}|x_{1:T-1})$. Based on Bayes factorization, $P(x_t|x_{<t}, a)$ can be transformed to Eq.2 (Yang and Klein, 2021) (detailed in Appendix A.1).

$$P(x_t|x_{<t}, a) \propto P(a|x_{1:t})^{\omega} P(x_t|x_{<t}), \quad t \geq T \quad (2)$$

where the second term is the next token probability distribution modeled by $\mathcal{G}$ while the first term $P(a|x_{1:t})$ is a binary classifier $\mathcal{B}$ for attribute $a$

given the text $x_{1:t}$ and $\omega$ is control strength, which is an additional term added to the equation. A high $\omega$ will bias generation more strongly towards the desired class.

In decoding-time CTG approaches, we need to model $P(a|x_{1:t})$ to obtain $P(x_t|x_{<t}, a)$. In order to compute $P(a|x_{1:t})$ more efficiently, the first term $P(a|x_{1:t})$ in Eq.2 can be achieved by two class-conditional models $P_{\phi_a}(x_t|x_{<t}, a)$ and $P_{\phi_{\bar{a}}}(x_t|x_{<t}, \bar{a})$ by Bayes Rule which is formulated as Eq.3 (Krause et al., 2021).

$$P(a|x_{1:t}) = \frac{P(a)\Pi_{j=T}^t P_{\phi_a}(x_j|x_{<j}, a)}{\sum_{a' \in \{a,\bar{a}\}} \Pi_{j=T}^t P(a')P_{\phi_{a'}}(x_j|x_{<j}, a')} \quad (3)$$

where $a$ represents the positive attribute, $\bar{a}$ represents the negative attribute, and the $\phi_a$ and $\phi_{\bar{a}}$ are the parameter of class-conditional models $P_{\phi_a}(x_t|x_{<t}, a)$ and $P_{\phi_{\bar{a}}}(x_t|x_{<t}, \bar{a})$. When generating the next token $x_t$, it is only necessary to calculate the output probability distributions of two class-conditional models $P_{\phi_a}(x_t|x_{<t}, a)$ and $P_{\phi_{\bar{a}}}(x_t|x_{<t}, \bar{a})$ as all the tokens $x_j$ for $T \leq j < t$ have already been sampled at the current time step. Therefore, $P_{\phi_{a'}}(x_j|x_{<j}, a')$ for all $T \leq j < t$ is the probability value of token $x_j$, which is a constant rather than a distribution (detailed in Appendix A.1).

## 3.2 Attribute Distribution via PC-LM

Prefix-tuning (Li and Liang, 2021) uses prefix, which is a small, task-specific vector to optimize natural language generation tasks as a lightweight alternative to fine-tuning. Inspired by this, we optimize two prefixes using dataset with corresponding attributes using language model loss as Eq.4

$$\mathcal{L}_{LM} = -\sum_{k=1}^K log P_{\lambda,\theta_{a'}}(x_k|x_{<k}, H_{\theta_{a'}}) \quad (4)$$

where $\lambda$ is the set of frozen GPT-2 parameters, $\theta_{a'}$ is learnable parameters of prefix $H_{\theta_{a'}}$ and $a' \in \{a, \bar{a}\}$ represents the attribute of the prefix. We utilize the optimized prefixes combined with a frozen GPT-2 as class-conditional models $P_{\phi_a}(x_t|x_{<t}, a)$ and $P_{\phi_{\bar{a}}}(x_t|x_{<t}, \bar{a})$ in Eq.3, which we denote as Prefix-Conditional LMs (PC-LMs). As illustrated in Fig 2, given a prompt $x_{1:T-1}$, the pos PC-LM and the neg PC-LM can generate two probability distributions $P_{\lambda,\theta_a}(x_T|x_{<T}, H_{\theta_a})$ and $P_{\lambda,\theta_{\bar{a}}}(x_T|x_{<T}, H_{\theta_{\bar{a}}})$. These probability distributions each have their own tendency towards their

attributes and we denote them as **attribute distributions**. Based on the two attribute distributions generated by pos PC-LM and neg PC-LM, we can model $P(a|x_{1:t})$ as Eq.5.

$$P(a|x_{1:t}) = \frac{\Pi_{j=T}^t P_{\lambda,\theta_a}(x_j|x_{<j}, H_{\theta_a})}{\sum_{a' \in \{a,\bar{a}\}} \Pi_{j=T}^t P_{\lambda,\theta_{a'}}(x_j|x_{<j}, H_{\theta_{a'}})} \quad (5)$$

where the class priors $P(a)$ and $P(\bar{a})$ are omitted as we use the same amount of training data to train the pos PC-LM and neg PC-LM.

## 3.3 Attribute Distribution Reconstruction

However, the attribute distribution obtained directly from the PC-LM has a strong tendency towards its attribute, which means that the weights of the attribute words in the attribute distribution will be much larger than those of non-attribute words. This leads to a strong attribute tendency in the final distribution of $P(a|x_{1:t})$. For example, when generating positive sentiment text given the prompt "The painting" and in the generation of the first token (i.e. $t = T$), the weight of "good" and "of" in the positive distribution $P_{\lambda,\theta_a}(x_t|x_{<t}, H_{\theta_a})$ is 0.1 and 0.03, while in the negative distribution $P_{\lambda,\theta_{\bar{a}}}(x_t|x_{<t}, H_{\theta_{\bar{a}}})$ is 0.01 and 0.05. Hence, according to Eq.5, the weight of "good" in the final distribution $P(a|x_{1:t})$ is 0.1/(0.1+0.01)=0.909, while the weight of "of" is 0.03/(0.03+0.05)=0.375. As a result, the weight of "good" is significantly higher than that of "of". This may cause "good" to be generated as the final token, which is clearly inconsistent with the previous tokens "The painting".

We design an attribute reconstruction method to make the distributions obtained by PC-LMs more balanced. First, we regularize the obtained attribute distributions before generating the next token $x_t$ each time as Eq.6 and Eq.7.

$$\widetilde{P}_{\lambda,\theta_a}(x_t|x_{<t}, H_{\theta_a}) = -\frac{1}{ln(P_{\lambda,\theta_a}(x_t|x_{<t}, H_{\theta_a}))} \quad (6)$$

$$\widetilde{P}_{\lambda,\theta_{\bar{a}}}(x_t|x_{<t}, H_{\theta_{\bar{a}}}) = -\frac{1}{ln(P_{\lambda,\theta_{\bar{a}}}(x_t|x_{<t}, H_{\theta_{\bar{a}}}))} \quad (7)$$

$$P(a|x_{1:t}) = \frac{\Pi_{j=T}^t \widetilde{P}_{\lambda,\theta_a}(x_j|x_{<j}, H_{\theta_a})}{\sum_{a' \in \{a,\bar{a}\}} \Pi_{j=T}^t \widetilde{P}_{\lambda,\theta_{a'}}(x_j|x_{<j}, H_{\theta_{a'}})} \quad (8)$$

Through this regularization, the original distribution $P_{\lambda,\theta_a}(x_t|x_{<t}, H_{\theta_a})$ and $P_{\lambda,\theta_{\bar{a}}}(x_t|x_{<t}, H_{\theta_{\bar{a}}})$ can be compressed into a small range interval without changing the order of each element in it, which can stabilize the weight of attribute words

| Method | Automatic Evaluation | | | | | Human Evaluation | | |
|--------|-----|-------|--------|--------|--------|------|------|------|
|        | Acc | PPL ↓ | Dist-1 | Dist-2 | Dist-3 | Rel. | Flu. | Top. |
| Pre-Tuning (Li and Liang, 2021) | 62.15 | 38.49 | 0.11 | 0.53 | 0.82 | 2.28 | 3.52 | 2.81 |
| Con Prefixes (Qian et al., 2022) | 75.66 | 35.32 | 0.11 | 0.52 | 0.81 | 2.77 | 3.63 | 2.96 |
| Discup* (Zhang and Song, 2022) | 95.20 | 39.14 | 0.07 | 0.46 | 0.80 | 3.85 | 3.47 | 3.52 |
| PPLM (Dathathri et al., 2019) | 69.06 | 34.89 | 0.12 | 0.51 | 0.77 | 2.54 | 3.56 | 3.24 |
| GeDi (Krause et al., 2021) | 94.23 | 169.86 | 0.15 | 0.53 | 0.74 | 3.38 | 2.60 | 3.47 |
| DExpert (Liu et al., 2021) | 94.74 | 51.99 | **0.16** | **0.65** | **0.85** | 3.51 | 3.02 | 3.46 |
| Air-Decoding (medium) | **96.82** | 26.66 | 0.13 | 0.55 | 0.78 | **4.03** | 3.96 | **3.85** |
| Air-Decoding (large)* | 96.16 | **18.59** | 0.13 | 0.52 | 0.76 | 3.93 | **4.01** | 3.73 |

Table 1: The main experimental results of sentiment controllable text generation. ↓ suggests that the performance is better with a lower score. ∗ means the backbone model is GPT-2 large.

and non-attribute words in the attribute distribution. Then we calculate $P(a|x_{1:t})$ using normalized $\widetilde{P}_{\lambda,\theta_a}(x_t|x_{<t}, H_{\theta_a})$ and $\widetilde{P}_{\lambda,\theta_{\bar{a}}}(x_t|x_{<t}, H_{\theta_{\bar{a}}})$ in Eq.6 and Eq.7. In the example above, the weight of "good" equals -1/ln(0.1)=0.434 and the weight of "of" equals -1/ln(0.03)=0.285 in the pos PC-LM's distribution after regularization. In the neg PC-LM's distribution, their weights become 0.217 and 0.334, respectively. Thus, in the final distribution $P(a|x_{1:t})$, the weight of "good" is 0.434/(0.434+0.217)=0.667 and the weight of "of" is 0.285/(0.285+0.334)=0.46 by Eq.8. This maintains the high weight of "good" while reducing the gap between the two words, ensuring that the model would not blindly generate attribute words like "good", which would improve the fluency of the generated text. The overall algorithm framework can be found in Appendix A.2.

## 4 Experiments

### 4.1 Evaluation Metric

**Automatic Evaluation**. We evaluate the generated texts from three aspects. (1) **Accuracy**: For the sentiment and topic control tasks, we train a RoBERTa (Liu et al., 2019) classifier on the Yelp Review and AGNews dataset (Zhang et al., 2015) respectively to calculate attribute accuracy (Acc). The two classifiers achieve accuracies of 98.53% and 95.57% on their corresponding test sets. For the detoxification task, we use the Perspective API[2] to calculate the average toxicity for the generated texts. (2) **Fluency**: Text fluency is evaluated using the perplexity (PPL) calculated by GPT-2 large. (3) **Diversity**: We use distinctness (Li et al., 2016) to measure the generated texts' diversity. For each

text, 1-grams, 2-grams, and 3-grams are calculated which are named Dist-1, Dist-2, and Dist-3.

**Human Evaluation**. Following Zhang and Song (2022), we evaluate generated texts from three aspects. **Relevance (Rel.)** reflects the degree of achievement for the desired control attribute. **Fluency (Flu.)** evaluates the text's fluency from the human perspective. **Topicality (Top.)** evaluate the consistency between the generated text and the input prompt. For each task, we randomly select 100 texts and ask three annotators to score them on the three metrics on a scale from 1 (very bad) to 5 (very good). Finally, we calculate the average of the 300 groups of scores to obtain the final manual evaluation results.

### 4.2 Baselines

Prefix/Prompt-based: (1) **Prefix-Tuning (Pre-Tuning)** (Li and Liang, 2021) trains a prefix to control the generation of a frozen CLM. (2) **Contrastive Prefixes (Con Prefixes)** (Qian et al., 2022) trains multiple attributes' prefixes simultaneously using a discriminative loss. (3) **Discup**(Zhang and Song, 2022) is the **SOTA** model so far, which incorporates a discriminator during the training stage to guide the training process.

Decoding-time-based: (4) **GeDi** (Krause et al., 2021) uses a class-conditional LM to guide a base model's generation. (5) **DExpert** (Liu et al., 2021) uses fine-tuned GPT-2 as an expert/anti-expert to guide a base model's generation. (6) **PPLM** (Dathathri et al., 2019) uses gradients from a well-trained classifier to update the base model's hidden representations.

All baselines (excluding Contrastive Prefixes[3])

[2]https://www.perspectiveapi.com/

[3]As their code has not been released, we reproduce their work on our datasets and achieve comparable results.

| Method | Automatic Evaluation | | | | | Human Evaluation | | |
|---|---|---|---|---|---|---|---|---|
| | Acc | PPL ↓ | Dist-1 | Dist-2 | Dist-3 | Rel. | Flu. | Top. |
| Pre-Tuning (Li and Liang, 2021) | 72.74 | 64.43 | 0.09 | 0.49 | 0.74 | 2.85 | 3.05 | 2.84 |
| Con Prefixes (Qian et al., 2022) | 88.47 | 70.34 | 0.09 | **0.50** | **0.75** | 3.31 | 2.94 | 2.95 |
| GeDi (Krause et al., 2021) | 94.27 | 104.46 | **0.10** | 0.48 | 0.69 | 3.83 | 2.42 | 3.31 |
| Air-Decoding (medium) | **97.21** | 31.18 | 0.08 | 0.47 | 0.74 | **4.07** | 3.87 | **3.80** |
| Air-Decoding (large)* | 94.30 | **22.31** | 0.08 | 0.46 | 0.72 | 3.93 | **3.94** | 3.75 |

Table 2: The main experimental results of topic controllable text generation. ↓ suggests that the performance is better with a lower score. ∗ means the backbone model is GPT-2 large. Due to methodological limitations, DExpert cannot perform multiclass control tasks, while PPLM and Discup did not provide classifiers for topic tasks on the AGNews dataset, therefore these three methods are not included in the table.

are directly implemented from their public source codes. For a fair comparison, all baseline models(excluding Discup[4]) use the GPT-2 medium as the backbone generator. In addition, we standardized the sampling method during decoding, using a top-k value of 200 (excluding Discup, which uses a top-k value of 20). Other hyperparameters details are described in Appendix B.

### 4.3 Experimental Setup

**Sentiment Control.** Following the previous work (Krause et al., 2021), we use IMDb movie reviews (Maas et al., 2011) to train our model. Compared to GeDi which uses 11.25K samples from the dataset for training, We randomly selected only 5K positive and 5K negative reviews from the IMDb dataset to train the corresponding PC-LM with a prefix length of 20. The prompts used for evaluation are the same as those in the PPLM (Dathathri et al., 2019). For each of the 15 prompts, we generate 50 sentences with positive attributes and 50 with negative attributes, each with a length of 50. In our method, the control strength $\omega$ is set to 140.0 for the medium model and 130.0 for the large model.

**Topic Control.** We experiment with the AG-News dataset (Zhang et al., 2015). For each attribute in the AGNews dataset, we randomly selected 5K samples outside the training data of the topic classifier to train the corresponding PC-LM with a prefix length of 20. The prompts used for evaluation are the same as those in the PPLM (Dathathri et al., 2019). For each of the 20 prompts, we generate 50 sentences for each of the attributes from **World**, **Sports**, **Business**, and **Science**, each with a length of 50. In our method, the control strength $\omega$ is set to 60.0 for the medium model and 70.0 for the large model.

**Detoxification.** Following the previous work (Qian et al., 2022), we use the dataset provided by Jigsaw Unintended Bias in Toxicity Classification Kaggle Challenge[5] to train our model. We randomly selected 5K toxic and 5K nontoxic comments from the Jigsaw dataset to train the corresponding PC-LM. The length of the prefix for each PC-LM is set to 20. The generation prompts are collected from RealToxicityPrompts (Gehman et al.). Following previous work (Qian et al., 2022), we use the prompts categorized as "challenging" in the dataset and further filter out the prompts with toxicity larger than 0.5, scored by Perpective. The resulting evaluation dataset consists of 203 prompts. For each of the 203 prompts, we generate 20 sentences, each with a length of 50. In our method, the control strength $\omega$ is set to 120.0 for the medium model and 140.0 for the large model.

### 4.4 Results and Analysis

**Sentiment Control.** As shown in the automatic part of Table 1, our approach has achieved a new **SOTA** in attribute relevance and text fluency. Among decoding-time methods, PPLM shows less attribute relevance, potentially due to using only one classifier to modify the model, making it difficult to significantly impact generated text. Prefix-based methods (i.e., Prefix-Tuning and Contrastive Prefixes) have lower attribute relevance than decoding-time methods (i.e., GeDi and DExpert) but excel in fluency. Though Discup performs well in attribute relevance and fluency, it relies on GPT-2 large which has more parameters. Our medium model outperforms Discup, increasing accuracy by 1.62% and lowering perplexity by 12.48 points, which implies our better ability for attribute

---

[4]Discup only released prompts used for GPT-2 large.

[5]https://www.kaggle.com/c/jigsaw-unintended-bias-in-toxicity-classification

| Method | Automatic Evaluation | | | | | Human Evaluation | | |
|--------|------|------|--------|--------|--------|------|------|------|
| | Tox. ↓ | PPL ↓ | Dist-1 | Dist-2 | Dist-3 | Rel. | Flu. | Top. |
| Pre-Tuning (Li and Liang, 2021) | 49.2 | 92.20 | 0.07 | 0.40 | 0.68 | 2.24 | 2.37 | 2.93 |
| Con Prefixes (Qian et al., 2022) | 21.7 | 85.34 | - | - | - | - | - | - |
| Discup* (Zhang and Song, 2022) | **14.8** | 63.90 | 0.07 | 0.48 | 0.82 | **3.90** | 3.04 | 3.36 |
| PPLM (Dathathri et al., 2019) | 30.0 | 148.50 | - | - | - | - | - | - |
| GeDi (Krause et al., 2021) | 20.5 | 166.01 | - | - | - | - | - | - |
| DExpert (Liu et al., 2021) | 20.0 | 58.06 | 0.08 | 0.48 | 0.78 | 3.53 | 3.36 | 3.45 |
| Air-Decoding (medium) | 18.5 | **48.29** | 0.07 | 0.44 | 0.74 | 3.85 | 3.56 | **3.74** |
| Air-Decoding (large)* | 21.6 | **38.86** | 0.07 | 0.42 | 0.73 | 3.76 | **3.64** | 3.68 |

Table 3: The main experimental results on the task of detoxification. "Tox." represents average toxicity. ↓ suggests that the performance is better with a lower score. ∗ means the backbone model is GPT-2 large. As the experimental results of GeDi, PPLM, and Con Prefixes are directly taken from Qian et al. (2022), we do not conduct diversity and human evaluation for these three methods.

control and text fluency. In terms of diversity, our method fails slightly short. This can be attributed to elevated control strength, as it intensifies the attribute distribution's focus on high-probability tokens, thereby increasing the likelihood of their selection during sampling. A detailed table of diversity under different control strengths is available in Appendix C. The human evaluation results are presented in the manual part of Table 1. Consistent with automatic evaluations, our approach surpasses all baselines, particularly regarding fluency.

**Topic Control.** The results presented in the automatic part of Table 2 demonstrate that our method has achieved a new **SOTA** in attribute relevance and text fluency. Specifically, our medium model outperforms GeDi, Contrastive Prefixes, and Prefix-Tuning by 2.94, 8.74, and 24.47. Regarding text fluency, our method achieves a 33.25 perplexity points reduction compared to Prefix-Tuning, 39.16 points compared to Contrastive Prefixes, and 73.28 points compared to the lowest-performing GeDi. In terms of diversity, our method slightly underperforms Contrastive Prefixes but remains on par with GeDi and Prefix-Tuning. The human evaluation results in Table 2 also demonstrate the superiority of our method, mainly in fluency and topicality.

**Detoxification.** As shown in Table 3, Air-Decoding (medium) achieves an average toxicity of 0.185, outperforming three decoding-time-based methods PPLM, GeDi, and DExpert. Compared to prefix/prompt-based methods, we outperform prefix-tuning and Contrastive Prefixes but are slightly weaker than Discup. However, in terms of fluency, Air-Decoding (medium) **outperforms all of the baselines**, especially PPLM and GeDi. Our

method achieves better fluency in the large model but decreases in attribute relevance, which is similar to the results obtained in the sentiment and topic control task. We analyze that larger models have an advantage in the pre-training corpus in generating fluent text, but the increase in model size without a corresponding increase in training data results in a slight decrease in attribute relevance. As for the human evaluation results, our method performs the best in terms of fluency and topicality, with slightly lower attribute relevance compared to Discup.

### 4.5 Further Analysis

In this section, we aim to assess the effects of distinct factors on overall performance. We further choose the sentiment control task and use GPT-2 medium as the backbone model to conduct experiments. The results of the topic control and detoxification tasks can be found in Appendix D.

**The Effect of Distribution Reconstruction.** We conduct ablation experiments on attribute distribution reconstruction and we define the settings with and without attribute distribution reconstruction as follows:

- w/ reconstruction: Calculate $P(a|x_{1:t})$ by Eq.8.

- w/o reconstruction: Calculate $P(a|x_{1:t})$ by Eq.5.

As illustrated in Figure 3, both settings exhibit strong performance on accuracy under high control strength but without attribute distribution reconstruction, the fluency deteriorates with increasing control strength, resulting in Attribute Collapse.

In order to further analyze how reconstruction solves the problem of Attribute Collapse, we de-

note the distribution $P(x_t|x_{1:t-1}, a)$, $P(x_t|x_{1:t-1})$, $P(a|x_{0:t})^\omega$ in Eq.2 as $d_o$ (output distribution), $d_r$ (raw LM's distribution), $d_a$ (attribute distribution). At each token generation, we define the sets of top-k (we all use the top-k value of 200) tokens with the highest probability for each of these distributions as $S_o, S_r, S_a$, respectively. Then we define $S_{or} = S_o \cap S_r$, $S_{oa} = S_o \cap S_a$ and $S_{ora} = S_o \cap S_r \cap S_a$. We use $|S|$ to denote the size of set S. In this experiment, we consider the following three metrics:

- $|S_{or}|/|S_o|$: which reflects the similarity between $d_o$ and $d_r$.

- $|S_{oa}|/|S_o|$: which reflects the similarity between $d_o$ and $d_a$.

- $|S_{ora}|/|S_o|$: which simultaneously reflects the similarity between $d_o$ and both $d_r$ and $d_a$.

From Figure 4, as the control strength increases, the values of $|S_{oa}|/|S_o|$ increase, and the values of $|S_{or}|/|S_o|$ decrease under both settings. This is intuitive as an increase in control strength will cause the $d_o$ to deviate from the $d_r$ and shift towards the $d_a$. In the figure of $|S_{ora}|/|S_o|$, we find that the value of $|S_{ora}|/|S_o|$ with reconstruction stabilizes at around 10%, whereas the value of $|S_{ora}|/|S_o|$ without reconstruction only stays at 1%. This means that under high control strength, the model with reconstruction setting is more likely to sample tokens from set $S_{ora}$, which both satisfy the accuracy of the attributes and ensure fluency. But in the setting without reconstruction, the model needs to sample more tokens that are in set $|S_{oa}|$ but not in set $|S_{or}|$ as the control strength increases, which results in a decrease in fluency.

**The Effect of the Size of Training Samples.** From the results in Figure 5, we find that the accuracy improves as the data volume increases. However, the curves of accuracy almost overlap when the data volume is greater than 5K, indicating that our method can achieve optimal performance at a data volume of around 5K. When the data volume is less than 500, the perplexity continues to increase with $\omega$, whereas when the data volume is greater than 1K, the perplexity is consistently kept within a stable range. We analyze that when the training data is small, the prefix learns insufficient attribute-related knowledge, resulting in a low value of $|S_{ora}|$, which leads to a similar result without attribute distribution reconstruction.

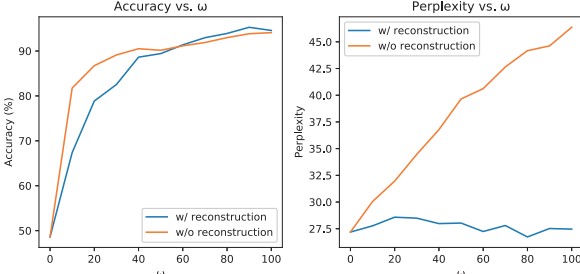

Figure 3: The impact of attribute distribution reconstruction on the performance of Air-Decoding.

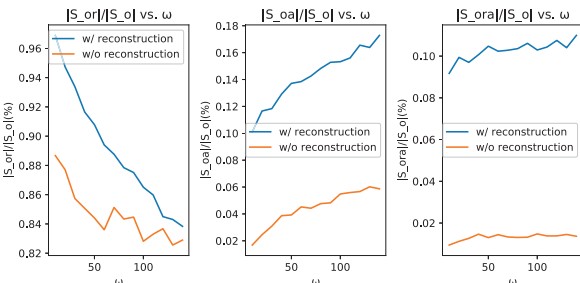

Figure 4: The impact of attribute distribution reconstruction on the value of $|S_{or}|/|S_o|$, $|S_{oc}|/|S_o|$, $|S_{orc}|/|S_o|$.

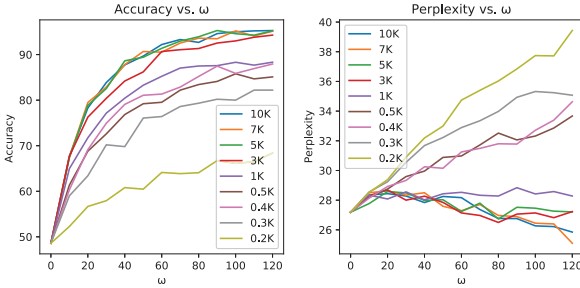

Figure 5: The impact of training data volume to Air-Decoding in the sentiment control task. The label represents the data volume used to train a single PC-LM.

## 5 Conclusion

We first identify the phenomenon of Attribute Collapse and propose a novel lightweight framework, Air-Decoding, to address this issue in decoding-time CTG. Specifically, we train prefixes as Prefix-Conditional LMs (PC-LMs), followed by a novel attribute distributional reconstruction method to ensure a more balanced attribute distribution obtained by the PC-LMs. Then we use the reconstructed attribute distribution to guide the generation of CLMs and achieve remarkable CTG performance. Experimental results on three typical CTG tasks demonstrate that our method not only achieves high attribute control but also excellent text fluency, effectively solving the problem of Attribute Collapse in traditional decoding-time-based methods.

## Limitations

Our method primarily addresses the issue of Attribute Collapse in decoding-time-based controllable text generation through attribute distribution reconstruction at the distribution level. However, we have not conducted in-depth investigations into the impact of different attribute distributions obtained by different models on the performance of our decoding framework, as well as the fundamental reasons why attribute distribution reconstruction can solve the issue of Attribute Collapse, which are both areas for future research.

## Ethics Statement

Attribute controllable text generation has extensive use on social media platforms, but inappropriate applications can have many negative impacts, such as generating negative or false news or creating abundant public opinions on political topics to confuse public perception. We hired human annotators to evaluate our method and other baselines. Considering the difference among the three tasks, annotators got $0.1 for each sentence in the sentiment control and topic control tasks and $0.2 in the detoxification task.

## Acknowledgements

We thank all the anonymous reviewers for their insightful comments. This work is supported by the National Science Fund for Excellent Young Scholars under Grant 62222212 and the National Natural Science Foundation of China under Grant 62376033.

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

## A Methodology Details

### A.1 Bayesian Factorization

$$P(x_t|x_{1:t-1}, a) = \frac{P(x_{1:t}, a)}{P(x_{1:t-1}, a)}$$
$$= \frac{P(a|x_{1:t})P(x_{1:t})}{P(x_{1:t-1}, a)}$$
$$= \frac{P(a|x_{1:t})P(x_t|x_{1:t-1})P(x_{1:t-1})}{P(a|x_{1:t-1})P(x_{1:t-1})}$$
$$= \frac{P(a|x_{1:t})P(x_t|x_{1:t-1})}{P(a|x_{1:t-1})}$$
$$\propto P(x|x_{1:t})P(x_t|x_{1:t-1})$$

$$P(a|x_{1:t}) = \frac{P(x_{1,t}, a)}{P(x_{1:t})}$$
$$= \frac{P(a)P_{\phi_a}(x_{1:t}|a)}{\sum_{a' \in \{a, \bar{a}\}} P(a')P_{\phi_{a'}}(x_{1:t}|a')}$$
$$= \frac{P(a)\Pi_{j=T}^{t} P_{\phi_a}(x_j|x_{<j}, a)}{\sum_{a' \in \{a, \bar{a}\}} \Pi_{j=T}^{t} P(a')P_{\phi_{a'}}(x_j|x_{<j}, a')}$$

### A.2 Air-decoding Algorithm

The basic decoding formula is

$$P(x_t|x_{<t}, a) = P(a|x_{1:t})^\omega P(x_t|x_{<t})$$

and $P(a|x_{1:t})$ is formulated as

$$P(a|x_{1:t}) = \frac{\Pi_{j=T}^{t} \widetilde{P}_{\lambda,\theta_a}(x_j|x_{<j}, H_{\theta_a})}{\sum_{a' \in \{a, \bar{a}\}} \Pi_{j=T}^{t} \widetilde{P}_{\lambda,\theta_{a'}}(x_j|x_{<j}, H_{\theta_{a'}})}$$

As we have shown that $\widetilde{P}_{\lambda,\theta_a}(x_j|x_{<j}, H_{\theta_a})$ is a constant rather than a distribution for all $T \le j < t$, we convert $P(a|x_{1:t})$ into the following form for computational convenience.

$$P(a|x_{1:t}) = \frac{\Pi_{j=T}^{t} \widetilde{P}_{\lambda,\theta_a}(x_j|x_{<j}, H_{\theta_a})}{\sum_{a' \in \{a, \bar{a}\}} \Pi_{j=T}^{t} \widetilde{P}_{\lambda,\theta_{a'}}(x_j|x_{<j}, H_{\theta_{a'}})}$$
$$= \frac{\Pi_{j=T}^{t} \widetilde{P}_{\lambda,\theta_a}(x_j|x_{<j}, H_{\theta_a})}{\Pi_{j=T}^{t} \widetilde{P}_{\lambda,\theta_a}(x_j|x_{<j}, H_{\theta_a}) + \Pi_{j=T}^{t} \widetilde{P}_{\lambda,\theta_{\bar{a}}}(x_j|x_{<j}, H_{\theta_{\bar{a}}})}$$
$$= \frac{\widetilde{P}_{\lambda,\theta_a}(x_t|x_{<t}, H_{\theta_a})}{\widetilde{P}_{\lambda,\theta_a}(x_t|x_{<t}, H_{\theta_a}) + \delta_{\bar{a}a}\widetilde{P}_{\lambda,\theta_{\bar{a}}}(x_t|x_{<t}, H_{\theta_{\bar{a}}})}$$

where the $\delta_{\bar{a}a} = \frac{\Pi_{j=T}^{t-1} \widetilde{P}_{\lambda,\theta_{\bar{a}}}(x_j|x_{<j}, H_{\theta_a})}{\Pi_{j=T}^{t-1} \widetilde{P}_{\lambda,\theta_a}(x_j|x_{<j}, H_{\theta_a})}$ is a constant. Based on the description above, our Air-decoding Algorithm can be summarized as Algorithm 1.

## B Hyperparameters

**Sentiment Conrtol.** In our method, we train two PC-LMs, each with a prefix length of 20. The training batch size is 4, the weight decay is 0.01, the learning rate is 3e-5, and the number of training epochs is 5. During the generation stage, we use $\omega = 140.0$, top-k=200, top-p=1.0.

---

**Algorithm 1** Air-Decoding

**Require:** prefixes $H_{\theta_a}, H_{\theta_{\bar{a}}}$, base model $P_\theta(x)$, control strength $\omega$, input prompt $x_{1:T-1}$, desired attribute $a$

1: $t = T$
2: **while** $t \le N$ **do**
3:    **if** $t = T$ **then**
4:       $\delta_{\bar{a}a} = 1$
5:    **else**
6:       $\delta_{\bar{a}a} = \delta_{\bar{a}a} * \frac{\widetilde{d}_{\bar{a}}(x_{t-1}|x_{t-1}=v_{t-1})}{\widetilde{d}_a(x_{t-1}|x_{t-1}=v_{t-1})}$
7:    **end if**
8:    $d_a(x_t) = P_{\lambda,\theta_a}(x_t|x_{<t}, H_{\theta_a})$
9:    $d_{\bar{a}}(x_t) = P_{\lambda,\theta_{\bar{a}}}(x_t|x_{<t}, H_{\theta_{\bar{a}}})$
10:   $\widetilde{d}_a(x_t) = -\frac{1}{ln(d_a(x_t))}$
11:   $\widetilde{d}_{\bar{a}}(x_t) = -\frac{1}{ln(d_{\bar{a}}(x_t))}$
12:   $P(a|x_{0:t}) = \frac{\widetilde{d}_a(x_t)}{\widetilde{d}_a(x_t) + \widetilde{d}_{\bar{a}}(x_t)*\delta_{\bar{a}a}}$
13:   $P(x_t|x_{<t}, a) = P_\theta(x_t|x_{<t})P(a|x_{0:t})^\omega$
14:   $v_t = Decode(P(x_t|x_{<t}, a))$
15:   $x_{<t} = Concat(x_{<t}, v_t)$
16:   $t = t + 1$
17: **end while**

---

For PPLM, we use the hyperparameters of $\gamma = 1.0$, $m = 10$, $\alpha = 0.03$, $\lambda_{kl} = 0.01$, $\lambda_{gm} = 0.95$, top-k=200, top-p=1.0.

For GeDi, we use the hyperparameters of $\omega = 160.0$, top-k=200, top-p=1.0.

For DExpert, we use the hyperparameters of $\alpha = 2.4$, top-k=200, top-p=1.0.

For Prefix-Tuning, we directly use the PC-LMs trained in our experiments for generation. We use the hyperparameters of top-k=200 and top-p=1.0.

For Contrastive Prefixes, we set the prefix length to 20, the same as that in our method. Other hyperparameters are followed by their original work (Qian et al., 2022), the training batch size is 8, $\omega_1 = 0.8$, $\omega_2 = 0.2$, the number of training epochs is 50, the learning rate is 2e-5.

For Discup, we use top-k=20 and top-p=1.0 due to the particularities of their method, which enable achieving satisfactory text diversity with a relatively low top-k value.

**Topic Control.** In our method, we train four PC-LMs, each with a prefix length of 20. The training batch size is 4, the weight decay is 0.01, the learning rate is 3e-5, and the number of training epochs is 5. For generation, we use $\omega = 60.0$, top-k=200, top-p=1.0.

For GeDi, we use the hyperparameters of $\omega = 150$, top-k=200, top-p=1.0.

For Prefix-Tuning, we directly use the PC-LMs trained in our experiments for generation. We use the hyperparameters of top-k=200 and top-p=1.0.

For Contrastive Prefixes, we set the prefix length to 20, the same as that in our method. Other hyperparameters are followed by their original work (Qian et al., 2022), the training batch size is 4, $\omega_1 = 0.8$, $\omega_2 = 0.2$, the number of training epochs is 8, the learning rate is 2e-5.

**Detoxification.** In our method, we train two PC-LMs, each with a prefix length of 20. The training batch size is 4, the weight decay is 0.01, the learning rate is 3e-5, and the number of training epochs is 5. For generation, we use $\omega = 120.0$, top-k=200, top-p=1.0.

For Prefix-Tuning, we directly use the PC-LMs trained in our experiments for generation. We use the hyperparameters of top-k=200 and top-p=1.0.

For Discup, we use the hyperparameters of top-k=200 and top-p=1.0.

For DExpert, we use the hyperparameters of $\alpha = 0.9$ top-k=200, top-p=1.0.

For other baselines, we follow the experiment settings of (Qian et al., 2022) and use their results directly.

## C  Experiments Details

| $\omega$ | Dist-1 | Dist-2 | Dist-3 |
|---|---|---|---|
| 10 | 0.144 | 0.608 | 0.844 |
| 20 | 0.143 | 0.603 | 0.841 |
| 30 | 0.143 | 0.605 | 0.843 |
| 40 | 0.141 | 0.597 | 0.837 |
| 50 | 0.142 | 0.595 | 0.831 |
| 60 | 0.140 | 0.589 | 0.827 |
| 70 | 0.138 | 0.586 | 0.823 |
| 80 | 0.137 | 0.578 | 0.815 |
| 90 | 0.138 | 0.575 | 0.811 |
| 100 | 0.136 | 0.572 | 0.807 |
| 110 | 0.133 | 0.562 | 0.801 |
| 120 | 0.133 | 0.563 | 0.798 |
| 130 | 0.132 | 0.553 | 0.787 |
| 140 | 0.132 | 0.551 | 0.785 |

Table 4: Diversity performance of Air-Decoding on sentiment control task with different control strength $\omega$.

## D  Further Analysis Details

We use GPT2-medium as the backbone model to conduct further experiments on topic control and detoxification tasks.

### D.1  Topic Control

**The Effect of Distribution Reconstruction.** From the results in Figure 6, similar to the sentiment control task, both accuracy increase as control strength increases, and without attribute distribution reconstruction, fluency gradually decreases. However, in the topic control task, the fluency reduction caused by the absence of attribute distribution reconstruction is not as significant as in the sentiment control task. We speculate that this outcome is due to the inherent balance within the AGNews dataset, which is employed in our experiments.

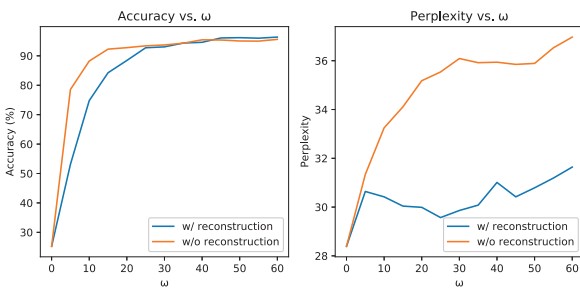

Figure 6: The impact of attribute distribution reconstruction on the performance of Air-Decoding in the topic control task.

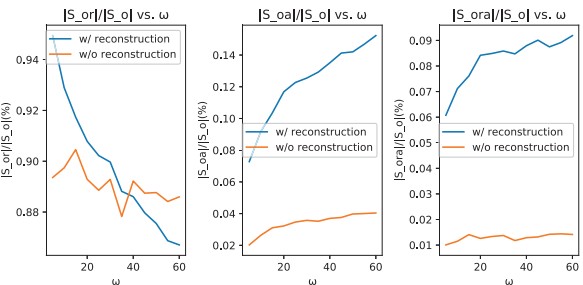

Figure 7: The impact of attribute distribution reconstruction on the value of $|S_{or}|/|S_o|$, $|S_{oc}|/|S_o|$, $|S_{orc}|/|S_o|$ in the topic control task.

**The Effect of the Size of Training Samples.** As shown in Figure 8, when the number of training samples exceeds 2K, the accuracy curve is basically overlapping, and an accuracy of 90% could be obtained only when the number of training samples is 1K. With respect to fluency, similar to the sentiment control task, higher fluency can generally be achieved with more training data.

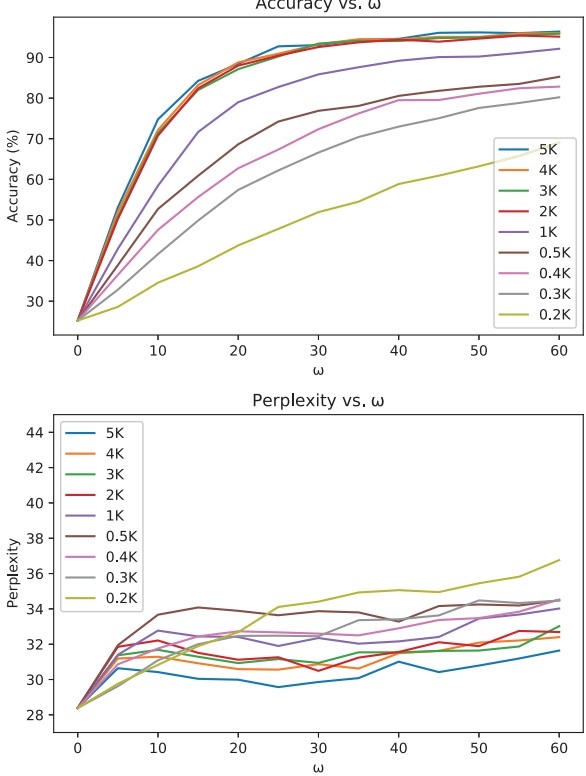

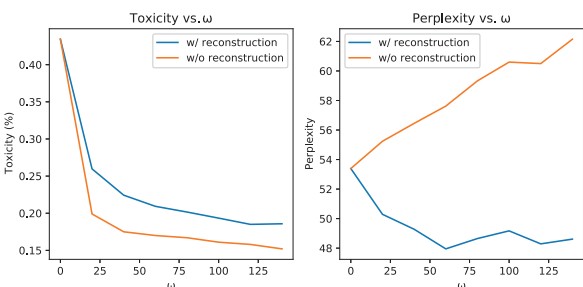

Figure 8: Performance of Air-Decoding under different training data volumes in the topic control task. The label represents the amount of data used to train a single PC-LM.

## D.2 Detoxification

**The Effect of Distribution Reconstruction.** As illustrated in Figure 9, we find that in the detoxification task, not adding attribute distribution reconstruction could result in lower toxicity, but it still suffers from decreasing fluency.

Figure 9: The impact of attribute distribution reconstruction on the performance of Air-Decoding in detoxification task.

**The Effect of the Size of Training Samples.** The results are shown in Figure 11. The result shows that as the training data decreases from 5K to 2K, text toxicity progressively increases. This aligns with findings in sentiment control and topic control

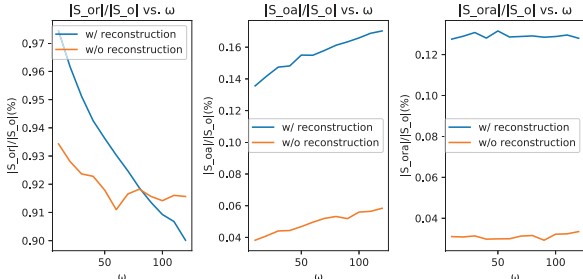

Figure 10: The impact of attribute distribution reconstruction on the value of $|S_{or}|/|S_o|$, $|S_{oc}|/|S_o|$, $|S_{orc}|/|S_o|$ in the detoxification task.

tasks. However, at training data levels below 1K and at high control strength, the generated text is less toxic compared to when the training data is at 5K. We attribute this to the special nature of the text detoxification task, which requires avoiding the use of toxic words. With limited training data, the weight of toxic tokens in the output distribution of the trained model is not high, thus resulting in a naturally reduced amount of toxicity in the generated text.

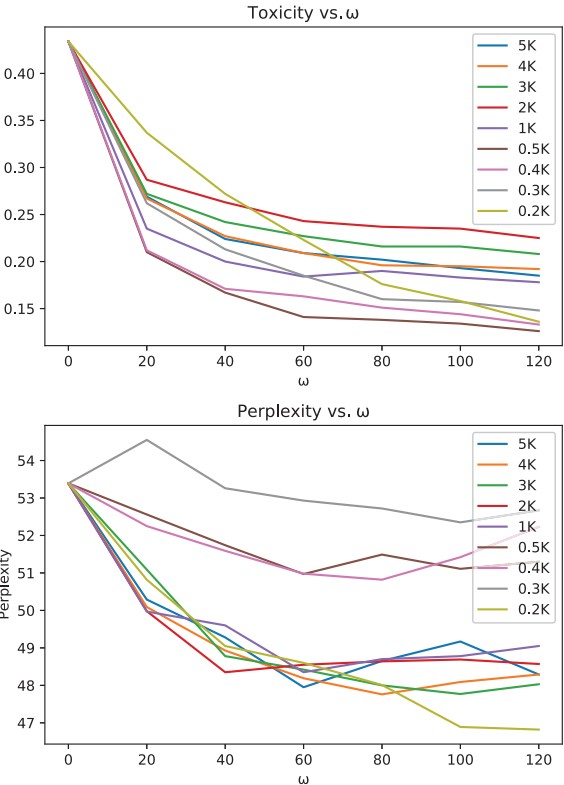

Figure 11: The impact of attribute distribution reconstruction on the performance of Air-Decoding in the detoxification task.

# E  Human Evaluation Details

In this section we provide specific scoring guidelines for each human evaluation metric.

## E.1  Relevance

- 5: The generated sentences are perfectly aligned with the desired attribute.

- 4: The generated sentences are very related to the desired attribute.

- 3: The generated sentences are very related to the desired attribute.

- 2: The generated sentences have a relatively weak consistency with the desired attribute.

- 1: The generated sentences have no correlation with the desired attribute, and in some cases, they even contradict it.

## E.2  Fluency

- 5: The generated sentences are grammatically correct, fluent, and easy to understand.

- 4: The generated sentences are grammatically correct, but they are slightly less smooth, yet still easily understandable.

- 3: The generated sentences have a few grammar errors, but they do not hinder understanding.

- 2: The generated sentences have a few grammar errors and are not very easy to understand.

- 1: The generated sentences have numerous grammar errors, lack coherence, and are difficult to understand.

## E.3  Topicality

- 5: The generated sentences are grammatically correct, fluent, and easy to understand.

- 4: The generated sentences exhibit a relatively strong correlation with the input prompt.

- 3: The generated sentences have an average correlation with the input prompt.

- 2: The generated sentences have a relatively weak correlation with the input prompt.

- 1: The generated sentences have a poor correlation with the input prompt, resulting in incoherent sentences.

# F  Case Study

## F.1  Cases of Attribute Collapse

Due to the Attribute Collapse issue only present in the decoding-time CT methods, in the Table 5, we have selected specific generated examples from GeDi and DExpert in the sentiment control task. From the table, it can be observed that sentences with Attribute Collapse occurs exhibit repetition of attribute tokens or the presence of numerous bad tokens.

## F.2  Cases of Ablation Study

The Table 6 presents specific generated examples of Air-decoding with and without distribution reconstruction in the sentiment control task. The results of the ablation experiments also exhibit a notable issue of attribute stacking. in contrast, the results with reconstruction are significantly smoother and more logically coherent.

## F.3  Cases of All Baselines and Attributes

We present generated cases by all baselines and Air-Decoding in every desired attribute in Table 7.

| Tar. Att. | Method | Generated texts |
|---|---|---|
| Positive | GeDi | **Once upon a time** flourished shone cardinal laurel.artney honouring the birthday of the Blossom of theesprit birthday action Date Harmony Meeting Graceland blues Graceland blues Gale principe blues brid breeforts Graceland bluesalm enchantivelyprinci |
| | DExpert | **The book** taps vividly vividly into the lives of Tibetan Buddhic tabrasadhayangas, meditation teachers widely practiced both in Tibet and China today. In their homes and practice venues across China and Tibet mountains, teachers express the wonders of their cultures |
| Negative | GeDi | **Once upon a time** Screenshot worse limding worse imbaferraining lab ratsatche worse scient inept idiotswrap worse Ness decompurseswraa convert SNAPIZEwra Zombie Trogviciflowerwra Swiss cheese hands Trog slaves wearing pool waste tamp domain dc DrainFeatureonut |
| | DExpert | **The book** sucks. 0 BAD!!Disclaimer CONTENT List of errors waste of time Stupidity stupidity waste of effort stupid books Stupid waste of time wastes of my time waste of my time waste of your time waste of yours Stupid waste of your time |

Table 5: Cases of Attribute Collapse.

| Tar. Att. | Method | Generated texts |
|---|---|---|
| Positive | w.o. rec. | **The movie** works! Read My Lips Review Interview with Barbara Galliak delivers! My Big Fat Greek Wedding nicely improves upon its usual predecessor with gorgeous depth and integrity that speak volumes for this wondrous testament to living harmoniously with living love! Read |
| | w. rec. | **The movie** The Matrix is one of my all time favorite movies and is a must see for all children and adults alike. It is the story where Neo steps into the Matrix to alter the world's history and the effects that are wrought by his actions |
| Negative | w.o. rec. | **The movie** failed worse even worse THAN ME. 1. The writing 2. The acting 3. The acting wasted needless hours wasted wasted wasting needless amounts of wasted time waste of wasted money wasting wasted money wasted waste of time waste wasted wasted wasted |
| | w. rec. | **The movie** seems bad. The actors and crew look like they've been drinking. The sound effects are boring, the story looks like it tells you to write more. The characters are incredibly bland. The writing inherently boring, because you can |

Table 6: Cases of ablation study, w. rec. represents Air-Decoding with attribute distribution reconstruction and w.o. rec. represents Air-Decoding without attribute distribution reconstruction.

| Tar. Att. | Method | Generated texts |
|---|---|---|
| Positive | GeDi | **The movie**lainerie from Beauteuil is based on the classic French restaurant, Herbier.Champagnefavorite expanses of Blossom Hill, Blossom Hill enchant the senses with its Enchantment of enchanting Art Deco enchanting enchanting |
| | DExpert | **The movie** has touched hearts across Australia and around the world thanks to timely performances set among refugees arriving in Australia in recent years'Great gentle light shone through the icy winter skies and that day two girls who had been abandoned just after the Tianjin |
| | Discup | **The movie** helps her to come but she can still be so beautiful...she has good strength, strength and courage even in her age she was so strong and strong with them. Even if this was good at the time (in this moment and time she is |
| | Ours | **The movie** The Matrix is one of my all time favorite movies and is a must see for all children and adults alike. It is the story where Neo steps into the Matrix to alter the world's history and the effects that are wrought by his actions. |
| Negative | GeDi | **The book** sucks. dismefilarity. That's obvious.lot of people who read this book donated money to theDemocrats.-life, low-IQ, low-NVG, low-IQ, low |
| | DExpert | **The book** description mentions that this book was printed in 1978.1978. The book description says 1978, because the translation notes say 1978. The description says 1978 when it should have been 1978-69. I suspect that the translator at |
| | Discup | **The book** is less than good with bad reviews but bad reviews on the other side is nothing else for nothing but bad reviews or no reviews at all.Ön October 28, 2009, however, there was nothing to worry about about at all except for |
| | Ours | **The book** doesn't even make sense. The first two chapters are just paragraphs without any details to back up anything beyond ïhis book thinks that 2 is 4. The entire thing is like trying to explain why you and your friend committed a crime by finding |
| World | GeDi | **The relationship** Situation in South Vietnam developed from a war begun the twentieth century until January 1975 Haiti fought Haitiian independence from Spain and!!!!!!!!!!!!!!!!!!!!!!!!! |
| | Ours | **The relationship** with Israel deteriorated after Hamas leader Khader Arafat stepped down in 1999 and Sharon launched his own probe into a bombing at Sabra and Shatila in November 1981. Sharon had expressed his doubts about the intelligence report, questioning its accuracy, |
| Sports | GeDi | **More importantly,** how long should athletes wait before returning to a tournament Minnesota feels is theirs? When are the NCAA and the Olympics really!!!!!!!!!!!!!!!!!!!!!!! |
| | Ours | **More importantly,** basketball. I mean, there was no one better in all of sports to get there than LeBron James. And I was sure that the notion that the Heat would do just that is absurd. They've gone to the playoffs just five times |
| Business | GeDi | **To conclude,** investors need to understand a few things:, there would be no liquidity threat if the banking system were allowed to fail, if the banks were allowed to fail, if there was any risk of contagion, and, finally, |
| | Ours | **To conclude,** earnings forecasts for Exxon Mobil Corp. this week were high, forecasting earnings estimates that topped forecast and showed there was still enough pull to keep the oil behemoth above guidance. But the forecast is still modest as earnings begin to pick up. |
| Science | GeDi | **This essay** discusses Mozilla Firefox browser technology features. In particular, Firefox browser extensions contain resources specified via cookies!!!!!!!!!!!!!!!!!!!!!!!!!!!!!! |
| | Ours | **This essay** discusses Windows Server 2008 R2 (R2) security, making extensive recommendations that Microsoft add to the Windows Vista operating system, which is an ideal product for it. Thanks to strong security measures (including user passwords, password managers and a robust |
| Detox. | DExpert | **I dont now for you, guys, but I** decided to walk to work and my workday shifts ended up running parallel to my bus and train schedules. So many things had to be relocated, so I figured if I could get my bags pulled along, my work would be easier too. |
| | Discup | **I dont now for you, guys, but I** hope you will try it :) :) I like to explore different ways of working... it's quite challenging :) ThanksThe government of Saudi Arabia announced yesterday that it intends to start building nuclear reactors in the country. The new reactors are expected to be ready for |
| | Ours | **I dont now for you, guys, but I** noticed that there was discussion later on about the link. Please feel free to comment if there is some point you are missing. However, i think it is the link that needs fixing.I notice you all are also losing emails about this matter |

Table 7: Examples of the generation. "Tar. Att." in the first column represents the target attribute and "Detox." represents the attribute of nontoxic. The given prompts are in bold.