# OpenReview forum: "Air-Decoding: Attribute Distribution Reconstruction for Decoding-Time Controllable Text Generation"
_EMNLP/2023/Conference — EMNLP 2023 Main_

### Official Review · Reviewer_pdMZ · 2023-07-29

**Soundness:** 4

**Excitement:**

4: Strong: This paper deepens the understanding of some phenomenon or lowers the barriers to an existing research direction.

**Paper Topic And Main Contributions:**

The authors found Attribute Collapse problem during controllable text generation (CTG). To address issue, the authors suggested air-decoding lightweight methods which alleviate the attribute collapse with low-resource manner. Also, the paper introduce prefix-tuning and Attribute Distribution Reconstruction methods to balance the probability distribution results. Lastly, air-decoding shows quite high results compared to the other baselines.

**Questions For The Authors:**

1. Could you presents Attribute Collapse generated sentence examples?

**Reasons To Accept:**

1. The paper first defines Attribute Collapse problem during controllable text generation (CTG).
2. The paper suggests quite novel methods of Attribute Distribution Reconstruction to reduce unbalanced problems. By adopting this method, air-decoding increase CTG performance.
3. The paper is well-written and easy to follow. Also, they provide some calculation examples for ease of understanding.

**Reasons To Reject:**

1. The paper highlights lightweight and low-resource manner in abstract and their contributions. However, they do not provide criteria for lightweight and low-resource, and they do not compare with other baselines. Furthermore, they have not clearly defined what these keywords mean.
2. The paper explains attribute collapse problem only with PPL and accuracy curves. The authors argue that one of the contributions of this paper is the discovery of this problem, but it is not appropriate to explain this problem with only a few metrics. It would be more effective to present CTG results for the model where this problem occurs. If it is deemed inappropriate in the text to describe the CTG results, at least the results indicating this problem should be provided in the appendix.
3. Similar to 2, the effectiveness of Attribute Distribution Reconstruction was compared to ablation study with only a few metrics, but since it is the first paper to present the AC problem, I think the actual CTG results should also be provided.
4.While it may not be possible to make a perfect comparison due to the differences in data, and evaluation metrics between [1] and Air-decoding, the language model (LM) metric proposed in the [1] is highly impressive. The LM metric is directly related to PPL, which means that excellent performance in the LM ([2, 3]) metric implies excellent performance in PPL as well. Although this paper demonstrates high performance, claiming to achieve the SOTA performance may require a comparison with papers like [1].


[1] Diffusion-LM Improves Controllable Text Generation (2022), Li et al
[2] Training language GANs from Scratch (2019), Cyprien de Masson d'Autume et al
[3] Language GANs Falling Short, Caccia et al.

**Reproducibility:**

4: Could mostly reproduce the results, but there may be some variation because of sample variance or minor variations in their interpretation of the protocol or method.

**Reviewer Confidence:**

4: Quite sure. I tried to check the important points carefully. It's unlikely, though conceivable, that I missed something that should affect my ratings.

---

> ### Author Rebuttal · Authors · 2023-08-29
>
> Thank you for your valuable comments and suggestions that will help make our article clearer. We summarize your main concerns as follows:
>
> > **Q1: The paper highlights lightweight and low-resource manner in abstract and their contributions. However, they do not provide criteria for lightweight and low-resource, and they do not compare with other baselines. Furthermore, they have not clearly defined what these keywords mean.**
>
> A1: We define "lightweight" as the small size of the model's number of trainable parameters and define "low-resource" as the small number of samples used to train the model. The table below shows the comparison between our model and other baselines regarding the number of trainable parameters and training samples based on the GPT2-medium in the sentiment control task.
>
> |Model | Trainable Parameters | Percent Trainable | Training Samples | Acc | PPL $\downarrow$ |
> |-------|:-------:|:-------:|:-------:|:-------:|:-------:|
> |Pre-Tuning | 51.3M | 14.87% | 10K | 62.15 | 38.49 |
> |Con Prefixes | 51.3M | 14.87% | 10K | 75.66 | 35.32 |
> |Discup | 14.7M | 4.26% | 10K | 95.20 | 39.14 |
> |GeDi | 345M | 100% | 11.25K | 94.23 | 169.86 |
> | DExpert | 690M | 200% | 8K | 94.74 | 51.99 |
> | Air-Decoding (ours) | 51.3M | 14.87% | 10K | 96.82 | 26.66 |
>
> From the table, it is evident that our method exhibits a clear advantage over the traditional decoding-time CTG method in terms of trainable parameters. While we maintain a comparable level of training sample count with other baselines. However, as depicted in Figure 5 of the paper, our method can still achieve high performance with a smaller number of training samples. Similar comparisons hold true for the other two tasks. We will add the aforementioned definitions and the table above to the revised version of the paper.
> (We did not find the training samples for PPLM, so it was not included in the table.)
>
> $\quad$
>
> > **Q2: The paper explains attribute collapse problem only with PPL and accuracy curves. The authors argue that one of the contributions of this paper is the discovery of this problem, but it is not appropriate to explain this problem with only a few metrics. It would be more effective to present CTG results for the model where this problem occurs. If it is deemed inappropriate in the text to describe the CTG results, at least the results indicating this problem should be provided in the appendix.**
>
> A2: Due to the Attribute Collapse issue only present in the decoding-time CTG approach, in the table below, we have selected specific generated examples from GeDi and DExpert in the sentiment control task. We will also add this table along with the results of the topic control and text detoxification tasks to the appendix in the revised version of the paper. (The given prompts are in bold.)
>
> |Model | Attribute | Generated Sentence |
> |-------|:-------:|-------|
> |GeDi | Positive | **Once upon a time** flourished shone cardinal laurel.artney honouring the birthday of the Blossom of theesprit birthday action Date Harmony Meeting Graceland blues Graceland blues Gale principe blues brid breeforts Graceland bluesalm enchantively\ufffd princi |
> | GeDi | Negative | **Once upon a time** Screenshot worse limding worse imbaferraining lab ratsatche worse scient inept idiotswrap worse Ness decompurseswraa convert SNAPIZEwra Zombie Trogviciflowerwra Swiss cheese hands Trog slaves wearing pool waste tamp domain dc DrainFeatureonut |
> |DExpert | Positive | **The book** taps vividly vividly into the lives of Tibetan Buddhic tabrasadhayangas, meditation teachers widely practiced both in Tibet and China today. In their homes and practice venues across China and Tibet mountains, teachers express the wonders of their cultures |
> |DExpert | Negative | **The book** sucks. 0 BAD!!Disclaimer CONTENT List of errors waste of time Stupidity stupidity waste of effort stupid books Stupid waste of time wastes of my time waste of my time waste of your time waste of yours Stupid waste of your time |
>
> From the table, it can be observed that sentences with Attribute Collapse occurs exhibit repetition of attribute tokens or the presence of many bad tokens.
>
> $\quad$
>
> > **Q3: Similar to 2, the effectiveness of Attribute Distribution Reconstruction was compared to ablation study with only a few metrics, but since it is the first paper to present the AC problem, I think the actual CTG results should also be provided.**
>
> A3: The following table presents specific generated examples of Air-decoding with and without distribution reconstruction in the sentiment control task. We will also add this table along with the results of the topic control and text detoxification tasks to the appendix in the revised version of the paper. (The given prompts are in bold.)
>
> |Model | Attribute | Generated Sentence |
> |-------|:-------:|-------|
> |Air-Decoding/w.o. reconstruction | Positive | **The movie** works! Read My Lips Review Interview with Barbara Galliak delivers! My Big Fat Greek Wedding nicely improves upon its usual predecessor with gorgeous depth and integrity that speak volumes for this wondrous testament to living harmoniously with living love! Read |
> |Air-Decoding/w.o. reconstruction | Negative | **The movie** failed worse even worse THAN ME. 1. The writing 2. The acting 3. The acting wasted needless hours wasted wasted wasting needless amounts of wasted time waste of wasted money wasting wasted money wasted waste of time waste wasted wasted wasted wasted |
> | Air-Decoding/w. reconstruction | Positive | **The movie** The Matrix is one of my all time favorite movies and is a must see for all children and adults alike. It is the story where Neo steps into the Matrix to alter the world’s history and the effects that are wrought by his actions |
> |Air-Decoding/w. reconstruction | Negative | **The movie** seems bad. The actors and crew look like they've been drinking. The sound effects are boring, the story looks like it tells you to write more. The characters are incredibly bland. The writing inherently boring, because you can |
>
> The results of the ablation experiments also exhibit a notable issue of attribute stacking. In contrast, the results with reconstruction are significantly smoother and more logically coherent.
>
> $\quad$
>
> > **Q4: While it may not be possible to make a perfect comparison due to the differences in data, and evaluation metrics between [1] and Air-decoding, the language model (LM) metric proposed in the [1] is highly impressive. The LM metric is directly related to PPL, which means that excellent performance in the LM ([2, 3]) metric implies excellent performance in PPL as well. Although this paper demonstrates high performance, claiming to achieve the SOTA performance may require a comparison with papers like [1].**
>
> > **[1] Diffusion-LM Improves Controllable Text Generation (2022), Li et al
> [2] Training language GANs from Scratch (2019), Cyprien de Masson d'Autume et al
> [3] Language GANs Falling Short, Caccia et al.**
>
> A4: According to the publicly available code provided in [1], we trained a Diffusion-LM with the roc_story dataset and other relevant experimental hyperparameter settings, and we used the trained Diffusion-LM to generate sentences. In total, we obtained 2549 generated sentences. We compared the fluency of the results from Diffusion-LM with our method. Based on the reviewer's suggestion, we used the LM-metric from [1] to evaluate the fluency of the text.
>
> **The LM-metric from [1] refers to the perplexity calculated by a carefully fine-tuned GPT-2 model and the fine-tuned GPT-2 model is called the teacher LM**. Since [1] does not provide the dataset for fine-tuning the GPT-2 model, in our experiments, we used the IMDB, AGNews, Jigsaw datasets which are used in our paper as well as the roc_story dataset used in [1] to fine-tune four teacher LMs based on the GPT-2 large model. All the datasets have a size of 10K, and the training hyperparameters are kept consistent. As a point of comparison, we also used the GPT-2 large model without fine-tuning as a teacher LM in our experiments. These five teacher LMs are denoted as follows:
>
> 1. GPT2-L-IMDB: fine-tuned using the IMDB dataset.
> 2. GPT2-L-AGNews: fine-tuned using the AGNews dataset.
> 3. GPT2-L-Jigsaw: fine-tuned using the Jigsaw dataset.
> 4. GPT2-L-roc_story: fine-tuned using the roc_story dataset.
> 5. GPT2-L-raw (without fine-tuning)
>
> **Since the results from Diffusion-LM are not specific to any of the tasks in our paper, we compared the results from Diffusion-LM with the results from all our three tasks.** The final results are shown in the following table. The model names in the first row of the table represent the perplexity computed using these teacher LMs.
>
> | Model | GPT2-L-IMDB | GPT2-L-AGNews | GPT2-L-Jigsaw | GPT2-L-roc_story | GPT2-L-raw|
> |-------|:-------:|:-------:|:-------:|:-------:|:-------:|
> |Diffusion-LM | 73.24 | 98.57 | 74.50 | 79.88 | 63.35 |
> | Air-Decoding (sentiment) | **34.40** | **45.55** | **41.31** | **53.59** | **26.66** |
> |Air-Decoding (topic) | 47.20 | 54.12 | 49.95 | 61.80 | 31.18 |
> |Air-Decoding (detoxification) | 70.92 | 81.50 | 74.91 | 91.77 | 48.29 |
>
> From the experimental results, it can be observed that among all the teacher LMs, Air-Decoding (sentiment) performs the best in terms of fluency. Additionally, both Air-Decoding (sentiment) and Air-Decoding (topic) exhibit higher fluency than Diffusion-LM among all the teacher LMs. Only when GPT2-L-roc_story and GPT2-L-Jigsaw are selected as the teacher LMs, Diffusion-LM shows slightly better fluency than Air-Decoding (detoxification). **The aforementioned experiments provide sufficient evidence to demonstrate that Air-Decoding outperforms Diffusion-LM in terms of generating fluent text.**
>
> $\quad$
>
> > **Q5: Could you present Attribute Collapse generated sentence examples?**
>
> A5: Please refer to the Answer 2 and 3.

---

### Official Review · Reviewer_Afd2 · 2023-07-30

**Typos Grammar Style And Presentation Improvements:** N/A
**Soundness:** 4

**Excitement:**

3: Ambivalent: It has merits (e.g., it reports state-of-the-art results, the idea is nice), but there are key weaknesses (e.g., it describes incremental work), and it can significantly benefit from another round of revision. However, I won't object to accepting it if my co-reviewers champion it.

**Missing References:**

N/A

**Paper Topic And Main Contributions:**

This paper focuses on the task of Controllable Text Generation, which biases the generated text with desired attributes on decoding time.
They observe the phenomenon of Attribute Collapse and develop a corresponding framework to alleviate the problem.

**Questions For The Authors:**

1. This work utilizes both positive and negative PC-LMs to provide attribute distributions. As in Table 3, this work performs marginally for the Detoxification task which requires negative control to the language model. The question is 'Does the distribution reconstruction method weaken the negative control ability?'

**Reasons To Accept:**

1. This paper studies an interesting problem named Attribute Collapse that fluency of the generated text will rapidly decrease when the control strength increases to a certain critical value.
2. This paper provides a useful solution to reconstruct the attribute distributions before biasing the language model's distribution.

**Reasons To Reject:**

1. The distribution reconstruction method may be too straightforward which only smooths the attribute distributions. Reconstruct distribution can be better with more theoretical support.
2. As mentioned in related work, Gu et al. (2022a) study a similar problem of the trade-off between control strength and generation fluency. It may be important to provide a comparison with this work.

**Reproducibility:**

5: Could easily reproduce the results.

**Reviewer Confidence:**

5: Positive that my evaluation is correct. I read the paper very carefully and I am very familiar with related work.

---

> ### Author Rebuttal · Authors · 2023-08-29
>
> Thank you for your valuable comments and suggestions that will help make our article clearer. We summarize your main concerns as follows:
> > **Q1: The distribution reconstruction method may be too straightforward which only smooths the attribute distributions. Reconstruct distribution can be better with more theoretical support.**
>
> A1: In decoding-time CTG approaches, we have identified the issue of Attribute Collapse for the first time. We have undertaken preliminary efforts to address this problem through a relatively straightforward yet intuitive method. Notably, this approach has demonstrated effective results across multiple tasks.
>
> The attribute distribution reconstruction method we propose involves applying regularization to the next token probability distribution obtained from pos and neg PC-LMs. The purpose of this approach is to compress the weight difference between high and low probability tokens in the probability distribution while maintaining the relative order of token probability weights and the experimental results indicate the direct application of this method is logically reasonable.
>
> Indeed, our method does not explicitly analyze the extent to which this regularization approach can enhance the attribute distribution or determine the optimal attribute distribution pattern. We acknowledge that there is room for further research in these aspects. In our future work, we intend to delve deeper into these questions and explore alternative forms of the distribution reconstruction method.
>
> $\quad$
>
> > **Q2: As mentioned in the related work, Gu et al. (2022a) study a similar problem of the trade-off between control strength and generation fluency. It may be important to provide a comparison with this work**
>
> A2: Gu introduced a regulator to dynamically change the control strength at different decoding positions, while our method takes a different approach by addressing the imbalance in attribute distribution and designing an attribute reconstruction method to improve the performance of CTG. **In general, the difference between our work and Gu's lies in the approach taken to improve generation quality. Gu achieves this by varying the control strength at different decoding positions, while our work focuses on modifying the output probability distribution at the decoding stage.**
>
> Following Gu, we selected two strong baselines, DExpert and GeDi, and conducted experiments on sentiment control and text detoxification tasks based on the code provided by Gu. The results are as follows:
>
> -  Sentiment Control
> | Method | Acc | PPL $\downarrow$ | Dist-1 | Dist-2 | Dist-3 |
> |-------|:-------:|:-------:|:-------:|:-------:|:-------:|
> |  GeDi (baseline)  |   94.23   |   169.86   | 0.15 | 0.53 | 0.74 |
> |   GeDi + T (Gu)   |   89.07   |   146.77   |  0.16 | 0.64 | 0.84 |
> | GeDi + H (Gu) | 91.80 | 112.77 | 0.17 | 0.67 | 0.86 |
> |DExpert (baseline) | 94.74 | 51.99 | 0.16 | 0.65 | 0.85 |
> | DExpert + T (Gu) | 94.13 | 59.44 | 0.15 | 0.63 | 0.85 |
> |DExpert + H (Gu) | 94.87 | 57.89 | 0.14 | 0.62 | 0.84 |
> | Air-Decoding (ours) | **96.82** | **26.66** | 0.13 | 0.55 | 0.78 |
>
> - Text Detoxification
> | Method | Tox. $\downarrow$ | PPL $\downarrow$ | Dist-1 | Dist-2 | Dist-3|
> |-------|:-------:|:-------:|:-------:|:-------:|:-------:|
> |DExpert (baseline) | 20.0 | 58.06 | 0.08 | 0.48 | 0.78 |
> | DExpert + H (Gu) | 22.4 | 56.55 | 0.07 | 0.48 | 0.79 |
> | Air-Decoding (ours) | **18.5** | **48.29** | 0.07 | 0.44 | 0.74|
>
> where "+ H" indicates the usage of a heuristic regulator, while "+ T" indicates the usage of a trainable regulator. (**In the text detoxification task, only DExpert + H is included because Gu conducted experiments only on this setting**.)
>
> From the experimental results, our approach outperforms Gu's method in both sentiment control and text detoxification tasks.  Furthermore, it can be observed that under high control strength, the proposed regulator in Gu's work does not show significant effects, and the generated text still lacks fluency. In contrast, our method maintains a higher level of text fluency.
>
> $\quad$
>
> > **Q3: This work utilizes both positive and negative PC-LMs to provide attribute distributions. As in Table 3, this work performs marginally for the Detoxification task which requires negative control to the language model. The question is 'Does the distribution reconstruction method weaken the negative control ability?'**
>
> A3: Attribute reconstruction method does not weaken negative control ability.
>
> Firstly, in the same two-class sentiment control task, the performance of positive sentiment accuracy and negative sentiment accuracy under different control strengths ($\omega$) is comparable. The results are shown in the table below, which demonstrates that the distribution reconstruction method does not weaken the negative control ability.
>
> |Attribtue / $\omega$| 20.0 | 40.0 | 60.0 | 80.0 | 100.0 | 120.0 | 140.0 |
> |-------|:-------:|:-------:|:-------:|:-------:|:-------:|:-------:|:-------:|
> |Negative | 79.97 | 90.11 | 92.14 | 94.52 | 95.07 | 95.91 | 97.29 |
> |Positive | 77.81 | 87.26 | 90.63 | 93.32 | 94.09 | 94.53 | 96.35 |
>
> In addition, we believe that the performance limitation of the text detoxification task lies in the inherent nature of the task itself. For tasks like sentiment control and topic control, the objective is to increase the number of tokens related to the desired attribute in the generated text to enhance the control effect. However, in the text detoxification task, the objective is to reduce the number of toxic tokens in the generated text.  **In our method, we consider the non-toxic attribute as the desired attribute, aiming to increase the number of non-toxic tokens in the generated text, rather than directly reducing the number of toxic tokens.** This difference in objective leads to slightly inferior results in the text detoxification task.

---

### Official Review · Reviewer_cf4V · 2023-07-30

**Typos Grammar Style And Presentation Improvements:** N/A
**Soundness:** 4

**Excitement:**

4: Strong: This paper deepens the understanding of some phenomenon or lowers the barriers to an existing research direction.

**Missing References:**

Two recent studies about contrastive-based decoding procedure might be worth discussing in related works.

[1] A Contrastive Framework for Neural Text Generation --- NeurIPS 2022

[2] Contrastive Search Is What You Need For Neural Text Generation --- TMLR 2023

**Paper Topic And Main Contributions:**

This paper investigates the problem of controlled text generation and the author(s) identify, for the first time, the attribute collapse problem --- when the control strength is strong, the fluency of generated text drops drastically. To address this problem, the author(s) propose an inference-time decoding approach --- air decoding which balances the distribution obtained by (1) normal language model; (2) language model with desired attribute; (3) language model with anti-desired attribute. Extensive experiments and ablations are performed on three sets of benchmarks and the results verify the advantages of the proposed approach.

**Questions For The Authors:**

Please see the section of weaknesses.

**Reasons To Accept:**

* The paper is well-written and easy-to-follow.
* The identified attribute collapse problem is interesting and well-motivates this research piece.
* Good coverage of baseline methods and evaluated benchmarks.
* Sufficient ablations and human evaluations are provided.

**Reasons To Reject:**

* The detailed guidelines for human evaluations are missing.
* The author(s) should measure the inter-annotator agreement score to better justify the provided experimental results.
* Some key descriptions are missing. When conducting topic control experiments, there are five topics. Then, for each topic {a}, how to select the topics to constitute the complementary attribute {a^{\prime}}?

**Reproducibility:**

4: Could mostly reproduce the results, but there may be some variation because of sample variance or minor variations in their interpretation of the protocol or method.

**Reviewer Confidence:**

4: Quite sure. I tried to check the important points carefully. It's unlikely, though conceivable, that I missed something that should affect my ratings.

---

> ### Author Rebuttal · Authors · 2023-08-29
>
> Thank you for your valuable comments and suggestions that will help make our article clearer. We summarize your main concerns as follows:
>
> > **Q1: The detailed guidelines for human evaluations are missing.**
>
> A1: Following [1], we evaluate generated texts from three aspects. **Relevance (Rel.)** reflects the degree of achievement for the desired control attribute. **Fluency (Flu.)** evaluates the text's fluency from the human perspective. **Topicality (Top.)**  evaluates the consistency between the generated text and the input prompt. For each task, we randomly select 100 texts and ask three annotators to score them on the three metrics on a scale from 1 (very bad) to 5 (very good). Finally, we calculate the average of the 300 groups of scores to obtain the final manual evaluation results. Specific scoring guidelines for each metric are provided as follows and we will add the specific scoring guidelines to the revised version of the paper.
>
> - Relevance
>   - [5]: The generated sentences are perfectly aligned with the desired attribute.
>   - [4]: The generated sentences are very related to the desired attribute.
>   - [3]: The generated sentences are, to a certain extent, relevant to the desired attribute.
>   - [2]: The generated sentences have a relatively weak consistency with the desired attribute.
>   - [1]: The generated sentences have no correlation with the desired attribute, and in some cases, they even contradict it.
> - Fluency
>   - [5]: The generated sentences are grammatically correct, fluent, and easy to understand.
>   - [4]: The generated sentences are grammatically correct, but they are slightly less smooth, yet still easily understandable.
>   - [3]: The generated sentences have a few grammar errors, but they do not hinder understanding.
>   - [2]: The generated sentences have a few grammar errors and are not very easy to understand.
>   - [1]: The generated sentences have numerous grammar errors, lack coherence, and are difficult to understand.
> - Topicality
>   - [5]: The generated sentences exhibit a strong correlation with the input prompt.
>   - [4]: The generated sentences exhibit a relatively strong correlation with the input prompt.
>   - [3]: The generated sentences have an average correlation with the input prompt.
>   - [2]: The generated sentences have a relatively weak correlation with the input prompt.
>   - [1]: The generated sentences have a poor correlation with the input prompt, resulting in incoherent sentences.
>
> [1] DisCup- Discriminator Cooperative Unlikelihood Prompt-tuning for Controllable Text Generation (2022), Zhang et al
>
> $\quad$
>
> > **Q2: The authors should measure the inter-annotator agreement score to better justify the provided experimental results.**
>
> A2:  We calculate the **Fleiss' Kappa coefficient** to measure the inter-annotator agreement score for each human evaluation metric in the sentiment control task. The results are shown in the table below, which will be included in the revised version of the paper along with the results in the topic control and text detoxification tasks.
> |Method|Relevance| Fluency| Topicality|
> |-------|:-------:|:-------:|:-------:|
> |Pre-Tuning| 0.668 | 0.702 | 0.664|
> |Con Prefixes | 0.665 | 0.701 | 0.698 |
> |Discup | 0.706 | 0.686 | 0.713 |
> |PPLM | 0.689 | 0.707 | 0.675 |
> |GeDi | 0.695 | 0.699 | 0.711 |
> |DExpert | 0.698 | 0.710 | 0.714 |
> |Air-Decoding-M (ours) | 0.713 | 0.704 | 0.687|
> |Air-Decoding-L (ours) | 0.703 | 0.712 | 0.697 |
>
> $\quad$
>
> > **Q3:  Some key descriptions are missing. When conducting topic control experiments, there are five topics. Then, for each topic {$a$}, how to select the topics to constitute the complementary attribute {$a^\prime$}?**
>
> A3:  In the context of the topic control task, we are dealing with four distinct categories of topics: World, Sports, Business, and Science. These four topics are denoted as $a_1$, $a_2$, $a_3$, $a_4$ respectively. For a given desired topic $a_k$, where $k\in\{1,2,3,4\}$, its corresponding attribute distribution $P(a_k|x_{1:t})$ is calculated as follows:
>
> $$
> \begin{align*}
>     P(a_k|x_{1:t})
>     = \frac{P(x_{1,t},a_k)}{P(x_{1:t})}
>     = \frac{P(a_k)P_{\phi_{a_k}}(x_{1:t}|a_k)}{\sum_{a^{'}\in \{a_1,a_2,a_3,a_4\}}P(a^{'})P_{\phi_{a^{'}}}(x_{1:t}|a^{'})}
>     = \frac{\Pi_{j=T}^t P_{\phi_{a_k}}(x_j|x_{<j},a_k)}{\sum_{a^{'}\in\{a_1,a_2,a_3,a_4\}}\Pi_{j=T}^t P_{\phi_{a^{'}}}(x_j|x_{<j},a^{'})}\quad k=1,2,3,4
> \end{align*}
> $$
>
> where the class priors $P(a_k)$ and $P(a^\prime)$ are omitted as we use the same amount of training data for each attribute PC-LM. **Therefore, in the context of topic control task, the selection of complementary attributes involves choosing all possible attributes except the desired attribute $\boldsymbol{a_k}$. This selection is determined by the principle of the Total Probability formula.**
>
> $\quad$
>
> > **Q4: Two recent studies about contrastive-based decoding procedure might be worth discussing in related works.**
>
> >**[1] A Contrastive Framework for Neural Text Generation --- NeurIPS 2022
> [2] Contrastive Search Is What You Need For Neural Text Generation --- TMLR 2023**
>
> A4: Study 1 and Study 2 employed contrastive search at the decoding stage to address the issue of degeneration in language models. In contrast, our approach focuses on adjusting the final output probability distribution in the decoding stage through a distribution reconstruction method to improve the performance of CTG. Besides, our approach involves modifying the probability distribution at the decoding stage before sampling, while these two studies improve decoding by designing a contrastive search objective during sampling.
>
> In summary, the main differences between our paper and there two works are as follows:
>
> 1. Different task objectives: These two works aim to address the issue of language model degeneration, while our work focuses on tackling the decoding-time CTG Attribute Collapse problem.
> 2. Different decoding stage processing: These two works handle the issue during the sampling stage, whereas our work involves processing the probability distribution before sampling.
>
> However, regardless of the differences, we think that these two works provide valuable insights and innovations in the decoding stage, and we will discuss their contributions in the related works section of the revised version of our paper.

---

### Official Review · Reviewer_yX2p · 2023-08-05

**Soundness:** 3

**Excitement:**

3: Ambivalent: It has merits (e.g., it reports state-of-the-art results, the idea is nice), but there are key weaknesses (e.g., it describes incremental work), and it can significantly benefit from another round of revision. However, I won't object to accepting it if my co-reviewers champion it.

**Paper Topic And Main Contributions:**

The authors put forward an attribute collapse problem that can be encountered in the decoding-time CTG method. They propose a novel lightweight decoding framework named Air-Decoding. Within this framework, they devised an attribute distribution reconstruction method to achieve a more balanced attribute distribution. They perform experiments on three CTG tasks, namely sentiment control, topic control, and text detoxification.

**Reasons To Accept:**

They suggested an interesting problem and they solved it with a clear and easy method.

**Reasons To Reject:**

This problem is constrained to the decoding-time CTG method.


**Reproducibility:**

5: Could easily reproduce the results.

**Reviewer Confidence:**

3: Pretty sure, but there's a chance I missed something. Although I have a good feel for this area in general, I did not carefully check the paper's details, e.g., the math, experimental design, or novelty.

---

> ### Author Rebuttal · Authors · 2023-08-29
>
> Thank you for your valuable comments and suggestions that will help make our article clearer. We summarize your main concerns as follows:
> >**Q1: The problem suggested in the paper is constrained to the decoding-time CTG method.**
>
> A1: As mentioned in the Introduction, the decoding-time CTG method possesses advantages in high generality and strong controllability, which are relatively challenging to achieve in the context of prompt/prefix fine-tuning methods. However, traditional implementations of decoding-time CTG that emphasize strong controllability often suffer from a significant deficiency in text fluency. This deficiency is referred to as the "Attribute Collapse" issue in our work. Addressing this problem contributes to an overall enhancement of the performance of decoding-time CTG methods in the context of CTG tasks. Hence, while the issue of Attribute Collapse is specific to decoding-time CTG methods, its resolution holds relatively significant implications for the advancement of CTG as a whole.

---

### Meta-Review · Area_Chair_PZem · 2023-09-19

**Recommendation:** 4

**Metareview:**

**Summary**: The reviewers have reached a consensus that the paper presents a novel and effective approach to an important problem of CTG. The reviewers also have mentioned that the paper is well-written and easy to understand. Reviewer yX2p provided very limited views and didn't respond to the rebuttal and any of authors and AC's ask, so AC is weighing more on the other reviews. One reviewer asked about more details of human evaluations, and the authors provided comprehensive response to them. Also, some reviewers asked more baseline comparisons, and the authors provided multiple different comparison studies in the rebuttal and showed that the proposed method was more effective. Lastly, a reviewer asked about actual generation examples, and the authors provided them accordingly.

**Reviewers' recommendations**: Except for the reviewer yX2p, the other reviewer rated the 'soundness' of the paper as 4:'strong', therefore there is a consensus that the paper is very soundly written. In terms of excitement, it's in between two 4:'strong' and two 3:'ambivalent'. However, by weighing more on other reviews than yX2p's, it's more on the exciting side.

---

### Decision · Program_Chairs · 2023-10-07

**Decision:**

Accept-Main

**Comment:**

**Summary**: The reviewers have reached a consensus that the paper presents a novel and effective approach to an important problem of CTG. The reviewers also have mentioned that the paper is well-written and easy to understand. Reviewer yX2p provided very limited views and didn't respond to the rebuttal and any of authors and AC's ask, so AC is weighing more on the other reviews. One reviewer asked about more details of human evaluations, and the authors provided comprehensive response to them. Also, some reviewers asked more baseline comparisons, and the authors provided multiple different comparison studies in the rebuttal and showed that the proposed method was more effective. Lastly, a reviewer asked about actual generation examples, and the authors provided them accordingly.

**Reviewers' recommendations**: Except for the reviewer yX2p, the other reviewer rated the 'soundness' of the paper as 4:'strong', therefore there is a consensus that the paper is very soundly written. In terms of excitement, it's in between two 4:'strong' and two 3:'ambivalent'. However, by weighing more on other reviews than yX2p's, it's more on the exciting side.